# Decentralized Zone-Based PKI: A Lightweight Security Framework for IoT Ecosystems

**Mohammed El-Hajj** *[ID] and **Pim Beune** [ID]

Department of Semantics, Cybersecurity & Services, University of Twente, 7522 NB Enschede, The Netherlands ; p.f.beune@alumnus.utwente.nl
* Correspondence: m.elhajj@utwente.nl

**Abstract:** The advent of Internet of Things (IoT) devices has revolutionized our daily routines, fostering interconnectedness and convenience. However, this interconnected network also presents significant security challenges concerning authentication and data integrity. Traditional security measures, such as Public Key Infrastructure (PKI), encounter limitations when applied to resource-constrained IoT devices. This paper proposes a novel decentralized PKI system tailored specifically for IoT environments to address these challenges. Our approach introduces a unique "zone" architecture overseen by zone masters, facilitating efficient certificate management within IoT clusters while reducing the risk of single points of failure. Furthermore, we prioritize the use of lightweight cryptographic techniques, including Elliptic Curve Cryptography (ECC), to optimize performance without compromising security. Through comprehensive evaluation and benchmarking, we demonstrate the effectiveness of our proposed solution in bolstering the security and efficiency of IoT ecosystems. This contribution underlines the critical need for innovative security solutions in IoT deployments and presents a scalable framework to meet the evolving demands of IoT environments.

**Keywords:** public key infrastructure; lightweight; Internet of Things (IoT); PKI; ECC; certificate; WSN; X.509

## 1. Introduction

The term Internet of Things (IoT) refers to tangible objects in the real world equipped with sensors, computing capabilities, and software, enabling them to communicate with other devices and systems via the Internet to exchange data [1]. Through wearables and personal mobile devices such as smartphones, the IoT is reshaping people's immediate environments into cyber-physical systems that they can interact with. For instance, various home automation devices enable consumers to automatically control household appliances, like turning off lights upon leaving home. Solutions focused on individuals greatly impact the daily lives of seniors and those with disabilities, enhancing their independence and confidence [2]. For example, voice-controlled devices empower visually or mobility-impaired individuals to manage home appliances [3]. IoT devices are inherently diverse [4] and resource-constrained [5], yet they are widely deployed globally, communicating wirelessly through protocols like Wi-Fi, Bluetooth, and Zigbee.

However, the IoT domain faces numerous security challenges that require attention [6]. Researchers have identified various attacks compromising the confidentiality [7–9], integrity [10–12], and availability [13,14] of IoT devices. For instance, Cui et al. [11] demonstrate how firmware upgrades can be exploited to introduce malicious alterations, leading to network spying and data breaches. The vast number of IoT devices presents an appealing target for adversaries [15] who may exploit them to create botnets, as illustrated by the Mirai botnet incident [16]. This botnet, comprising compromised IoT devices, orchestrated a significant DDoS attack in 2016, affecting numerous prominent websites [17].

Many of these security challenges can be mitigated through the implementation of cryptographic primitives, although it is crucial to ensure the proper implementation of

security measures, such as password authentication, to effectively safeguard IoT ecosystems. We shall now describe some of these cryptographic primitives.

- *Symmetric-key cryptography:* utilizes a single secret key for both encrypting and decrypting data. When the secret key is sufficiently robust, it becomes extremely difficult for an adversary, armed only with ciphertext, to uncover or alter the original plaintext. The Advanced Encryption Standard (AES) block cipher stands as a prominent illustration of symmetric-key encryption.
- *Public-key cryptography:* operates using pairs of interconnected keys, consisting of a public key and its corresponding private key for each pair. Securing public key cryptography necessitates keeping the private key confidential. With a public key, accessible to anyone, individuals can encrypt messages, generating ciphertext. However, solely those possessing the correlated private key can decrypt the ciphertext, unveiling the original message.
- *Public Key Infrastructure (PKI):* refers to a collection of software and hardware solutions designed to facilitate the management of public key encryption within computer systems. PKI aims to streamline the secure transfer of information over the Internet, particularly in scenarios where robust authentication is required, surpassing traditional methods like passwords. Its primary objectives include verifying the identities of parties involved in data transmission and ensuring the integrity of transferred data. PKI relies on digital certificates, which it can generate, distribute, modify, and invalidate. These certificates bind public keys to the identities, often referred to as "subjects", of various entities such as domain names. Certificate Authorities (CAs) play a crucial role in PKI by issuing and registering certificates as part of a comprehensive process that establishes these connections.

Figure 1 provides a concise outline of the PKI structure, consisting of five sequential steps:

**Step 1:** A server initiates the process by requesting a certificate from a Certificate Authority (CA). The server sends its Public Key (PK) and identity to the CA.

**Step 2:** Upon successful verification of the server's identity, the CA responds by issuing a certificate for the server's identity and PK. This certificate is signed with the CA's private key.

**Step 3:** A user wishes to establish a secure connection with the server. To achieve this, the user verifies the server's PK by requesting the server's certificate.

**Step 4:** In response to the user's request, the server provides the certificate.

**Step 5:** The user verifies that the certificate's signature indeed originates from the CA and confirms that the certificate contains the server's PK and identity. If these conditions are satisfied, the user trusts the server and proceeds to initiate a secure connection using the server's public key.

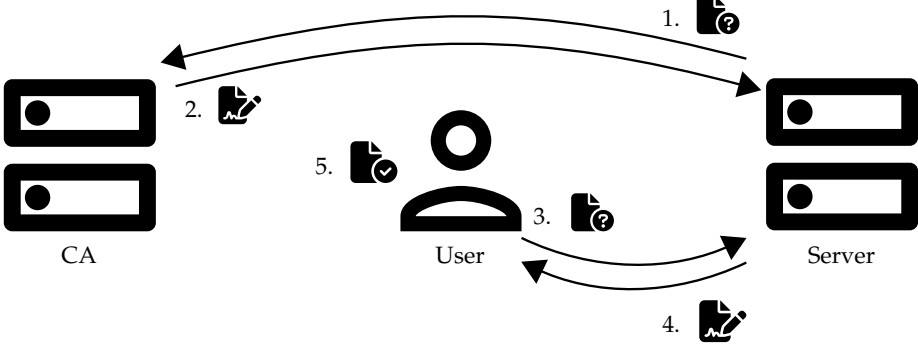

**Figure 1.** Concise overview of the PKI structure.

IoT vendors have been slow to adopt traditional Public Key Infrastructure (PKI) for several technical and financial reasons. Here are some key factors explaining why traditional PKI is not sufficient for the current IoT landscape:

1.  Traditional PKI architectures rely on Certificate Authorities (CAs), introducing a single point of failure. If attackers manage to compromise the CA, they can issue certificates for any entity, posing a significant threat in IoT environments. With the vast number of IoT devices, this creates a substantial attack surface [15].
2.  IoT devices typically have limited resources, which affects the performance of traditional cryptographic techniques. Research conducted by Blanc et al. [18] demonstrates significant performance discrepancies when executing identical encryption algorithms on various IoT hardware platforms compared to traditional PCs.

This paper endeavors to tackle the previously mentioned challenges by proposing a decentralized PKI system. To this end, we propose the following research questions:

- RQ1: How do we implement a lightweight PKI solution for the IoT?
- RQ2: How will this new lightweight PKI solution perform compared with traditional and literature PKI solutions?

The main contribution of this article is a PKI architecture that leverages lightweight cryptography and certificates, making it well-suited for IoT devices with limited computational capabilities. Furthermore, the design of this architecture reduces the threat of single point of failure by introducing so-called "zones", overseen by zone masters facilitating certificate management within its zone. The significance of this contribution is underlined by the points raised earlier, as well as the multitude of attacks documented against IoT devices in existing literature [7–15,19].

The structure of this article is as follows: Section 2 will briefly describe the systematic literature that has been undertaken beforehand. Then, Section 3 will describe the proposed PKI solution in detail. Next, Sections 4 and 5 will describe the experimental setup and results of the performance and security analysis, respectively. Finally, the paper will be concluded in Section 4.

## 2. Related Work

Drawing from insights gathered in a separate paper from the authors [20], which details the methodology and execution of our Systematic Literature Review (SLR) process, our proposed solution aims to bridge the gap between theory and practice in the realm of lightweight PKI for IoT. The SLR acted as our guide, navigating us through vast research from 2008 to 2023, ultimately highlighting 37 relevant studies out of 73 articles reviewed. While the details of our SLR journey are outlined in the aforementioned paper, our focus in this article is to translate those findings into a tangible framework for strengthening the security and efficiency of IoT ecosystems. To this end, we will now highlight three articles from the aforementioned article.

In Höglund et al. [5], the authors outline several obstacles to enabling PKI for the Internet of Things, as well as two solutions: certificate overhead reduction and secure enrollment. They do this by developing a new kind of X.509 certificate and shrinking its size by employing CBOR encoding. They have demonstrated their ability to securely complete initial enrollment and re-enrollment, and minimize X.509 overhead for the intended IoT applications. The authors demonstrate that the protocol uses around 4 KB of ROM and 1 KB of RAM. Furthermore, the compressed ECC certificates are only 150 bytes in size. Nevertheless, the proposed system does not protect itself against DoS attacks.

PKIoT is an architecture proposed by Marino et al. [21] with the goal of making certificate-based authentication practical for IoT devices with limited resources. The PKIoT architecture enables IoT nodes to delegate difficult security-related activities to a remote server. According to their present condition and level of trust in the server, nodes can freely choose which tasks to delegate. The PKIoT architecture provides an expandable, compatible, and flexible solution as a result. They also created a novel sort of compact certificate, which when used in place of standard X.509 certificates allows for even more reductions in transmission overheads but necessitates PKIoT support on both ends of the communication. Finally, their experimental results show that PKIoT is around 12×

faster in key generation and $10\times$ and $12\times$ faster in signature generation and verification, respectively, in comparison with not employing the PKIoT architecture.

A Lightweight Public Key Infrastructure called LPKI is presented in Toorani and Beheshti [22] and is tailored to platforms with limited resources. It utilizes signcryption and ECC, and gives each subscriber one set of private-public keys, delegating all validations to a trusted third party known as the validation authority. The architecture employs optimized certificates to increase protocol efficiency. Nevertheless, because the certificates have the same structure as the widely used X.509 certificates, they are easily compatible with the ubiquitous PKIX infrastructure. Unfortunately, neither security nor a performance analysis is present in this article.

## 3. Proposed Solution

The demand for a scalable and decentralized public key infrastructure (PKI) is on the rise as the Internet of Things (IoT) landscape expands at a rapid rate, connecting everything from sensors to appliances. A lightweight and decentralized PKI offers an appropriate solution to the particular security issues that IoT ecosystems raise.

Firstly, a PKI architecture for the IoT must be decentralized due to the nature of IoT networks. Decentralized PKIs reduce single points of failure and boost attack resistance simply by doing away with the need for a centralized authority to administer certificates. Devices may independently manage their cryptographic keys, create secure connections, and authenticate one another via a decentralized PKI, which promotes assurance within the IoT ecosystem.

Secondly, a lightweight PKI system is a PKI system created especially for IoT devices with limited resources. Such devices frequently feature little amounts of memory, computing power, and energy. The performance and efficiency of IoT devices can be hampered by traditional PKI structures, which were initially created for more powerful computing environments. IoT devices can use cryptographic processes that are tailored to their capabilities by implementing a lightweight PKI, which reduces processing overhead and energy usage while assuring reliable operation.

This section will describe the proposed solution for a novel PKI architecture tailored to IoT devices. The architecture hopes to address the limitations of current solutions presented earlier. The contributions of this proposed solution are two-fold:

1. In order to address the single point of failure present in traditional PKI architectures, we propose a decentralized PKI system. Previous literature has shown that this is feasible to achieve [4,23–29]. We propose to introduce an architecture that employs "zones", where each zone has a master (See Figure 2). This master issues, updates, revokes, and looks up certificates for all other IoT nodes in its zone. A node can become part of a zone by sending an enrollment request to a zone master, to which the master responds with a signed certificate (Figure 3). Thus, to a certain extent it acts as a CA when compared with the traditional PKI architecture. Furthermore, it is optimal for the master node to have significant computational power. The general idea is that zone masters keep track of the certificates of all nodes in its zone. If a zone gets too many nodes to handle, it will convert a node in the zone to a new master node, thereby creating a new zone, which is connected to the old zone. This creates a parent-child like architecture, where each zone (apart from the root zone) has a link to its parent and child zone. This zone system allows for high scalability, where each zone can represent a single entity. For example, one zone can contain all IoT nodes in a certain smart city, while another zone contains all nodes in a certain smart building.

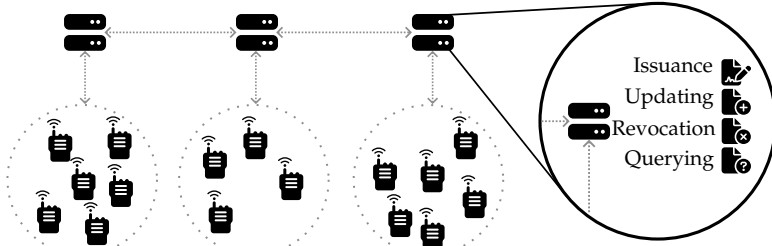

**Figure 2.** Overview of the proposed design. The gray circles represent zones, in which IoT devices are situated. The IoT devices are individually connected to their zone master located above them. Moreover, the zone masters can communicate with their neighbors.

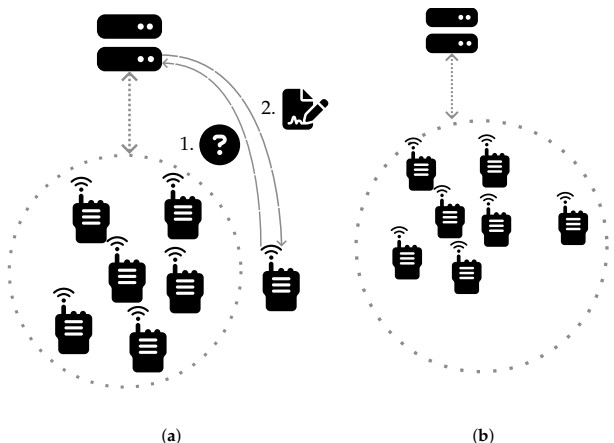

(**a**)　　　　　　　　(**b**)

**Figure 3.** Node enrollment procedure visualized. (**a**) A node wants to be enrolled into the zone, by sending an enrollment request to the zone master. The zone master sends back a signed certificate; (**b**) Next, the node is part of the zone.

2. The second contribution addresses the need for a lightweight architecture, which is achieved in two following ways:

   (a) The use of lightweight cryptography: Elliptic Curve Cryptography (ECC) is public key cryptography based on the algebraic structure of elliptic curves over finite fields. Because it offers the same level of security with smaller key sizes, it has an advantage over conventional cryptographic methods like RSA [30]. ECC starts with an elliptic curve that is described mathematically by the equation:

$$y^2 \equiv x^3 + ax + b \pmod{p} \tag{1}$$

   where $x$ and $y$ are coordinates on the curve, $a$ and $b$ are constants that define the shape of the curve, and $p$ is a prime number representing the finite field. One cryptosystem that makes use of ECC is the Elliptic-Curve Discrete Logarithm Problem (ECDLP); given the curve in Equation (1) and a point $P$ on the curve, the ECDLP involves finding an integer $n$ such that $nP$ equals a specified point $Q$ on the curve. In mathematical notation, the ECDLP can be expressed as:

$$Q = nP$$

   The security of elliptic curve cryptography relies on the difficulty of solving the Elliptic-Curve Discrete Logarithm Problem (ECDLP). That is, given a curve, a base point $P$, and a resulting point $Q$, it is computationally hard to determine the integer $n$ such that $Q = nP$ [31].

   (b) The use of lightweight certificates: the proposed architecture will dramatically reduce the number and size of fields present in the conventional X.509 certificates in order to speed up certificate creation and verification. To further reduce

communication overhead, we will use Concise Binary Object Representation (CBOR) to encode the certificates. CBOR is a binary data serialization format that resembles JSON in some ways. It enables the faster transport of data objects with name-value pairs than JSON. This sacrifices human comprehension in favor of more rapid processing and transfer rates.

We shall now describe a number of important procedures in this architecture. In these descriptions, we sometimes make use of symbols. Definitions of the symbols can be found in Table 1. Finally, lifecycle diagrams of both nodes and zones can be found in Figure 4.

**Table 1.** Symbols and notations used in this section.

| Symbol | Description |
|---|---|
| $H$ | Hardware score of a node |
| $U$ | Cryptographically secure Universally Unique Identifier (UUID) |
| $C$ | Node certificate |
| $S$ | Signature of $C$ |
| $PK$ | Public Key |
| $SK$ | Secret Key |
| $R$ | Revocation status (0 = not revoked, 1 = revoked) |
| $\{M\}_k$ | Message $M$ is encrypted with key $k$ |

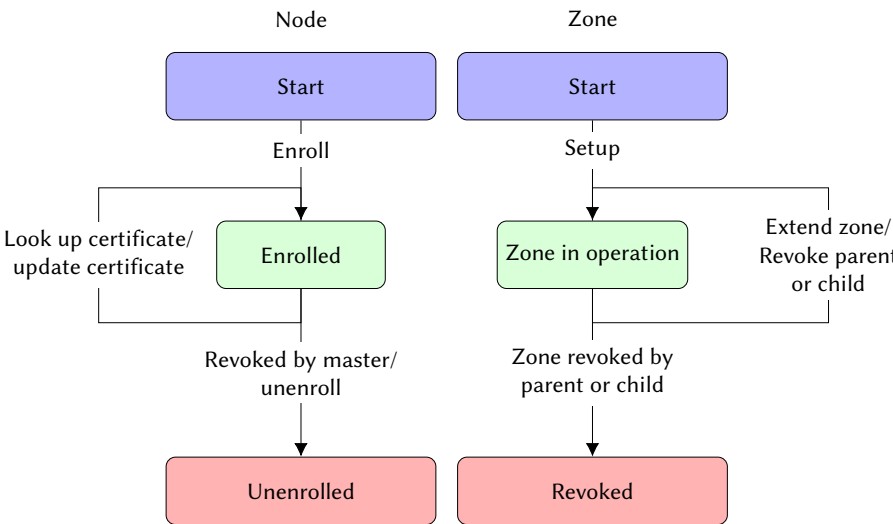

**Figure 4.** Lifecycle diagrams of a node and zone. The transitions between states contain hyperlinks to their respective sections.

### 3.1. Zone Setup

In order to set up a zone, a single zone master node must be deployed. Firstly, this zone master generates a cryptographically secure Public Key $PK$, Secret Key $SK$ and UUID $U$ for itself. The zone master will essentially act as a CA, compared with traditional PKI architectures. In order for the zone to be used by other nodes, its IP address must be published. When nodes obtain the IP address of a zone, they can enroll into the zone, which is described in the next section.

The zone that is set up first will be the zone whose IP address will be published for other nodes to join. This means that if a node does not know the IP address of a zone to join, it will try to join the first zone. After that, the zone master will decide to either welcome the node into its zone or to redirect its enrollment request to a child zone.

It is important to note that this operation is only applicable to zone masters who want to create a brand-new PKI chain. When a node wants to assist this PKI by becoming a zone master, it will first have to join the PKI as a regular node.

### 3.2. Node Enrollment

If a node wants to become part of a zone, it will first execute an Elliptic-Curve Diffie-Hellman (ECDH) key exchange so that subsequent communication can preserve confidentiality and integrity. The Diffie–Hellman key exchange technique enables two parties with no prior knowledge of one another to establish a common secret key across an insecure channel [32]. A symmetric-key cipher can then be employed with this key to encrypt subsequently sent messages. After the node and zone master have conducted a Diffie–Hellman key exchange to obtain a secure symmetric key $\sigma$, the node will use it to securely send an enrollment request to the zone master facilitated by the Advanced Encryption Standard (AES) block cipher. This enrollment request contains the hardware score $H$ of the node, which is calculated as follows:

$$H = \text{RAM in MB} \times \text{Weight}_{\text{RAM}} + \text{CPU Clock Speed in MHz} \times \text{Weight}_{\text{CPU Clock Speed}}$$
$$+ \text{Number of CPU Cores} \times \text{Weight}_{\text{CPU Cores}}$$

This hardware score incorporates various measurable properties of the hardware of a device. The three weights are determined beforehand by the zone master.

Optionally, depending on the level of trust in the zone master, the node will also generate a keypair itself and send the public key along as well. Thus, the node sends the following tuple to the zone master, where square brackets indicate optional fields:

$$\lambda = \{([PK], H)\}_{\sigma}.$$

Next, the zone master generates a cryptographically secure UUID $U$, $PK$ and $SK$ (if the node did not send along a public key), and certificate $C$. The certificate contains the UUID $U$ and public key $PK$, among others (Section 3.11). The certificate is signed with the $SK$ of the zone master, and stored in signature $S$. Moreover, it sends the tuple

$$\mu = \{([PK, SK], C, S)\}_{\sigma}$$

back to the node. Finally, the zone master stores the tuple

$$\nu = (U, H, C, R = 0)$$

in an internal table, where $R$ is the revocation status of the certificate. Because the certificate contains (among others) the public key of the node, the public key can be queried by other nodes if they want to securely communicate. We want to note that the zone enrollment relies on the principle of Trust on First Use (TOFU) [33]. That is, the zone master assumes the node to be benign and to not spoof their identity (UUID $U$) upon key generation. Moreover, we want to point out that after enrollment, all communication between zones and nodes is performed through public key cryptography (ECC) to ensure secure and private communication. An overview of the enrollment procedure is given in Figure 5.

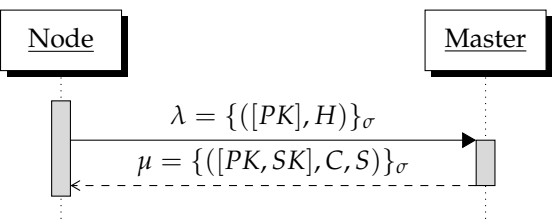

**Figure 5.** Node enrollment. Optional data are denoted by square brackets.

### 3.3. Subsequent Communication

After enrollment, the node can communicate with other nodes and its zone master using public key cryptography. Nevertheless, an adversary can perform a Man-in-the-

middle attack (MITM) attack, as illustrated in Figure 6. In this figure, a benign node requests the certificate of node $x$ to a benign zone master. The MITM, however, blocks the request from reaching the master and responds with a malicious response, encrypted with the public key of the node. Although the MITM still has to guess the type of request in order to successfully send back a malicious response, this is a significant problem.

To fix this problem, we propose that for every request, the requesting party sends along a nonce inside the encrypted message. Next, the responding party sends this same nonce inside the encrypted response. If the requesting party successfully verifies that the request nonce is identical to the response nonce, it can confirm that no MITM is present.

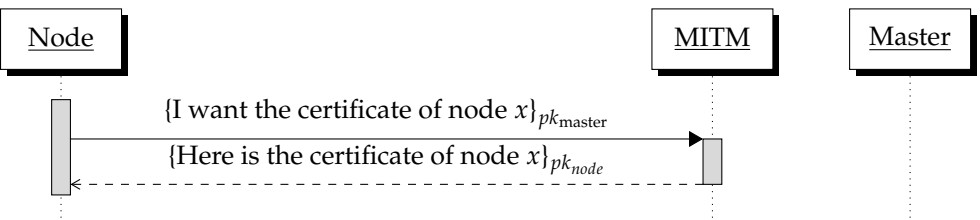

**Figure 6.** Demonstration of MITM attack.

The communication in the rest of this section will make use of this nonce principle.

### 3.4. Zone Extension

If a zone master does not want to accept any more node enrollments (e.g., if the zone master has too many nodes resulting in computational limits), it will instruct the strongest node, i.e., the node with the highest $H$ score in its zone, to create a new zone. We explain the next steps with the help of Figure 7. In the real architecture, nodes are identified by their UUID, but for readability purposes, we refer to them by a letter. Suppose zone master B wants to extend its zone, it will instruct the strongest node (C) to create a new zone (step 1). After this node (C) has created a new zone, the master of the old zone (B) will redirect the enrollment of new nodes to this new zone (C). Finally, the child (C) sends a recursive message to its parents (B and A) saying that it is the new child of B (step 2). Because zone master (A) knows master (B) is its child from the previous extension, it will respond to zone master (C) that it is C's grandparent (step 3). The grandparent/grandchild identification is performed in order to conduct zone master revocation, which is described in Section 3.9.

Due to accidental or purposeful misconfiguration, it can be possible that two zones redirect to each other upon enrollment of a node. When a node wants to enroll into a zone in such a case, it can become stuck in an infinite redirection loop. To solve this problem, we only allow zone masters to redirect enrollments to their child zones. Furthermore, when a zone master has vacant spots after a number of nodes have left, it can stop redirection upon enrollment of new nodes and welcome them into its zone.

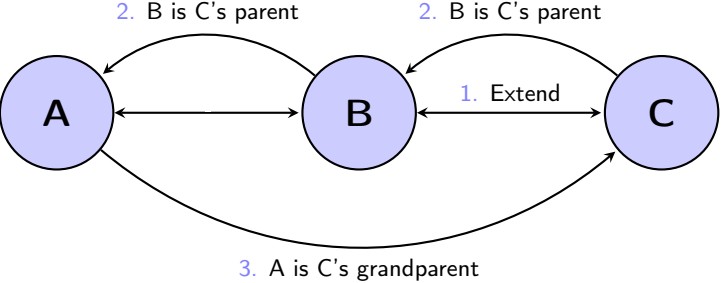

**Figure 7.** Zone extension procedure illustrated.

To conclude this section, we describe how a zone master determines whether it has reached its node limit, after which it will extend its zone. For performance purposes, we describe the following rule: if all nodes in a zone would be consecutively sending a request

to the zone master, the zone master must be able to complete all requests within one second. Next, we aim to determine this node threshold for a Raspberry Pi 4 Model B with 1 GB of RAM memory, and a Quad core 64-bit CPU @ 1.8 GHz, because this is the testbed for our performance analysis (Section 4). From this experiment, it is concluded that such a Raspberry Pi is able to handle 20 consecutive enrollment requests within one second. We decided to use the enrollment operation for this benchmark, as it requires a number of cryptographic operations, which would reflect a realistic situation in terms of computing power. Given that this Raspberry Pi has a hardware score of 5504 (with the three weights of the formula weighed equally), we deduced a formula to determine the maximum number of nodes per zone:

$$\text{Maximum number of nodes} = \text{Hardware score } (H) \times 3.6 \times 10^{-3}$$

### 3.5. Certificate Updating

It may be necessary for a node to update their public–private keypair, in case of private key loss or compromise. Should a node want to update its keypair, it can simply send a new certificate signed with the old key to its zone master. Then, the zone master will verify the signature, and if the signature is valid, it will replace the tuple $\mu$ in its internal table. In this tuple, $U$ and $H$ are copied from the previous entry, and $C$ is a new certificate of the node's new $PK$, signed with the $SK$ of the zone master.

### 3.6. Certificate Revocation

In case a node $U$ in a zone is acting maliciously, the zone master must be able to revoke the node's certificate. If this is the case, it can simply set the revocation status $R$ to 1 in its internal table. If other nodes want to look up the certificate of node $U$, the zone master will inform the nodes that $U$'s certificate has been revoked.

### 3.7. Certificate Lookup

When two nodes want to communicate securely, they must first obtain each other's certificates. When a node wants to look up the certificate $C$ of an identity $U$, it will ask its zone master for its certificate. The master is faced with two possible scenarios:

1. If $U$ is located in the master's zone, it can simply fetch the certificate from its internal table and send it back to the node.
2. If $U$ is not located in the master's zone, the procedure is as follows, accompanied by Figure 8: In this example, zone master 4 has the certificate for node 6. Node 5 asks its master (3) for the certificate of node 6. Because node 6 is not located in the zone of master 3, it will send a recursive query to its child and parent zone masters to ask if they have the certificate of 6. Zone master 4 responds with the certificate of node 6, and master 3 sends the certificate back to node 5.

Finally, when the node has obtained the requested certificate, it will verify all signatures of the zone masters until it reaches the root zone.

### 3.8. Certificate Verification

Before two nodes can communicate securely, the nodes must have verified each other's certificates. To this end, both nodes must first query the public key of each other's zone master. With this, they can verify each other's certificates. Next, both nodes must traverse the entire certificate chain up to the root zone by requesting the certificates and signatures of all zone masters that are a parent of the to-be-verified node.

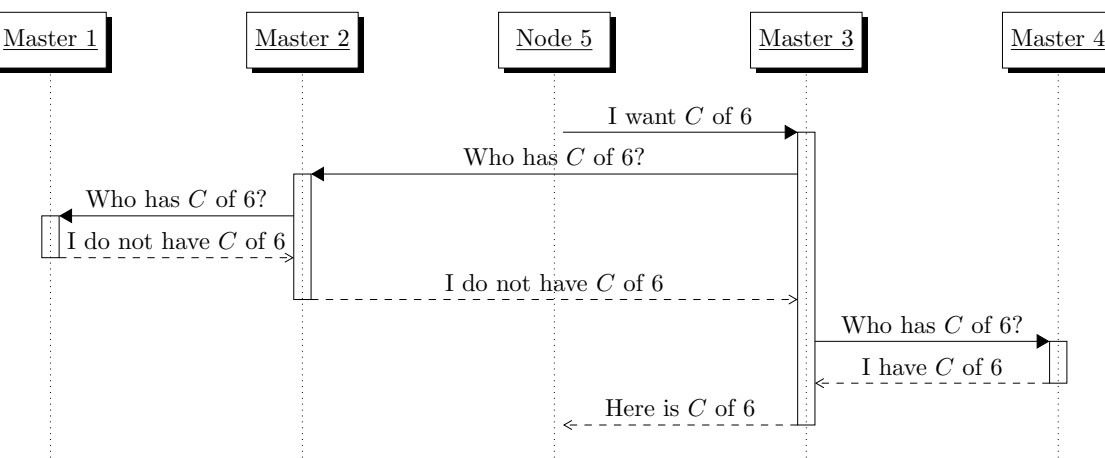

**Figure 8.** Certificate lookup procedure illustrated.

### 3.9. Zone Master Revocation

In case a zone master starts handing out malicious certificates, other zone masters must be able to remove the zone, i.e., revoke the zone. A zone master can revoke either its master parent or child. If a zone master decides to revoke its child zone master, its new child will become its grandchild zone master. If a zone master wants to revoke its parent zone master, the procedure is equivalent. Because the grandchildren and grandparents identify themselves in the zone extension procedure (Section 3.4), the master will know the identifiers $U$ of its grandchild and grandparent. Finally, the zone master will send a signed message to all nodes in the revoked zone that they have to unenroll and re-enroll so that they will be part of a valid zone again. An example is illustrated in Figure 9, where a zone master revokes its child zone.

Nevertheless, it is trivial that a malicious zone is not likely to update its nodes on the fact that it has been revoked. Therefore, the nodes that are in this revoked zone must be able to detect that its zone master has been revoked. To this end, we propose the following: periodically, all nodes in the zone will request proof from the parent and child zone master that they are still present. This will be completed in the form of a digital signature:

$$\text{sign}_{SK_P}(\alpha), \text{sign}_{SK_C}(\beta) \tag{2}$$

where $SK_P$ and $SK_C$ are the secret keys of the parent and child zones, and $\alpha$ and $\beta$ are random strings created by the verifying nodes in the zone. The only way for a malicious zone master to forge such signatures is to know the secret key of both zone masters. Should the parent and child zone masters return an incorrect or no signature, the node will assume that their zone has been revoked. After this, it will unenroll from the current zone and enroll in a new zone.

### 3.10. Node Unenrollment

When a node does not feel the necessity to communicate (securely) anymore, it can unenroll out of the zone. When a node wants to unenroll, it can simply send a signed request to the zone master, who will then revoke the node's certificate in its internal table, after successfully verifying the signature.

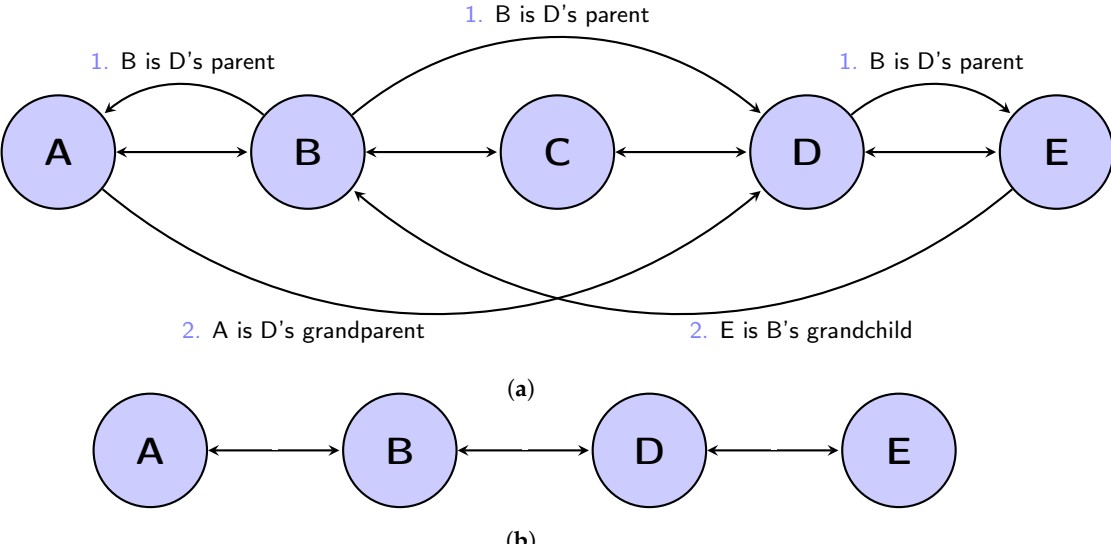

**Figure 9.** Zone revocation illustrated: all circles represent zone masters. In the real architecture, nodes are identified by their UUID, but for readability purposes, we refer to them by a letter. In this case, zone B revokes its child zone C (**a**), after which it is removed from the zone chain (**b**). (**a**) Zone B revokes its child zone C by telling all nodes (except C) that B is the new parent of D. Because other nodes know that B is their parent or child, they will update fellow nodes by telling them they are new grandparents/children; (**b**) while zone C can still operate, it is essentially removed from the zone chain.

### 3.11. Certificates and Keys

This section will describe a new certificate format that is based on X.509 certificates. In order to speed up certificate generation and verification, we will drastically reduce the number and size of fields found in the standard X.509 certificates, as similarly performed by Forsby et al. [34]. Moreover, we will encode the certificates using Concise Binary Object Representation (CBOR) to reduce the communication overhead even further. CBOR is a binary data serialization format that is somewhat related to JavaScript Object Notation (JSON). It permits the transfer of data objects containing name-value pairs (like JSON) but in a shorter way. This compromises human comprehension in favor of faster processing and transfer rates.

Finally, the key system to be used along with these certificates is ECC, as this is shown to be more efficient than other cryptosystems such as Rivest Shamir Adleman (RSA) [35–37] and has been widely implemented and tested on IoT devices [38–41]. Next, we shall describe the fields found in this new certificate architecture illustrated in Figure 10.

### 3.11.1. Version Number
X.509 Specification

This field is an integer describing the version of the certificate [42]. The versatility of X.509 version 3 offers support for other topologies like meshes and bridges [43]. Version 3 also offers support for certificate extensions, which allow a CA to exclusively issue certificates for predetermined uses.

```
TBSCertificate  ::=  SEQUENCE  {      Certificate  ::=  SEQUENCE  {
        version                               tbsCertificate
        serialNumber                          signatureAlgorithm
        signature                             signatureValue
        issuer                        }
        validity
        subject
        subjectPublicKeyInfo
        issuerUniqueID
        subjectUniqueID
        extensions
    }
```

**Figure 10.** Overview of a traditional X.509 certificate [42]. All fields are explained in the sections below.

Optimized Format

By restricting this field to only version 3, it can be left out entirely.

### 3.11.2. Serial Number
X.509 Specification

The serial number, which the CA assigns to each certificate, has to be a positive integer. It must be distinct for each certificate that a specific CA issues. In other words, the combination of CA and serial number uniquely identify a certificate [42].

Optimized Format

In order to reduce computational overhead as much as possible, the serial number will be a monotonically increasing integer, starting from 0.

### 3.11.3. Signature
X.509 Specification

The identifier for the algorithm that the CA employed to sign the certificate. The value of this field must be identical to the signature algorithm field (Section 3.11.11).

Optimized Format

By restricting the signature algorithm to a single algorithm, the field can be omitted entirely. The signature algorithm chosen is ECDSA NIST P-256, which is elaborated on in Section 3.11.7.

### 3.11.4. Issuer
X.509 Specification

The CA that issued the certificate is named in this field. The issuer field needs to have a Distinguished Name (DN) in it that is not empty [42].

Optimized Format

While the Issuer field in regular X.509 certificates contain various attributes such as country, organization, and Common Name (CN), in order to preserve space, this field shall only contain the CN.

### 3.11.5. Validity
X.509 Specification

The duration for which the CA guarantees that it will keep track of the certificate's state. The date the validity period starts (`notBefore`) and the date the validity period ends (`notAfter`) are expressed in the field as a sequence of two dates [42].

Optimized Format

The X.509 format allows for two date types:

1. UTCTime, which is of the format `YYMMDDHHMMSSZ`
2. GeneralizedTime, which is of the format `YYYYMMDDHHMMSSZ`

While the first date format is smaller, it does entail a drawback: the year is indicated using two characters. These two characters are interpreted as follows [42]:

- *"Where YY is greater than or equal to 50, the year SHALL be interpreted as 19YY; and"*
- *"Where YY is less than 50, the year SHALL be interpreted as 20YY."*

This means that after the year 2049, date types of this format are not applicable anymore. Nevertheless, the optimized format will still make use of this date type. In this optimized format, the YY characters are to be interpreted as 20YY.

### 3.11.6. Subject
X.509 Specification

The entity connected to the public key is identified by this field [42]. If the subject is a CA, the subject field must contain a "non-empty distinguished name matching the contents of the issuer field" [42].

Optimized Format

In order to provide unique identities for all nodes in the PKI, the subject will be a `uuid4`, with a length of 36 bytes.

### 3.11.7. Subject Public Key Info
X.509 Specification

This field contains the public key and specifies the algorithm that the key is used with.

Optimized Format

By allowing only one public key algorithm, the specified algorithm in this field can be omitted, thus decreasing computational overhead. The public key algorithm chosen is ECC, on the NIST P-256 curve. The reason for choosing Elliptic Curve Cryptography over a different asymmetric cryptosystem, e.g., RSA, is because it provides much higher levels of security for much smaller key sizes, thus reducing computational and communication overhead [30]. The P-256 curve is chosen because it provides 128 bit level security, which is considered 'acceptable' by the National Institute of Standards and Technology [30].

### 3.11.8. Unique Identifiers
X.509 Specification

To address the potential for subject and/or issuer name reuse. However, it is advised that CAs adopt distinctive names and must thereafter refrain from assigning these fields [42].

Optimized Format

Because it is advised that subjects and/or issuer names are not reused, this field is not necessary and therefore omitted entirely.

### 3.11.9. Extensions
X.509 Specification

This field lists one or more certificate extensions. It must allow at least the following extensions [42]:

- Key usage: This extension specifies the function of the key present in the certificate (such as certificate signing or encryption).
- Certificate policies: A collection of one or more policy information terms that describe the policy by which the certificate was issued and the permitted uses of the certificate.

- Subject alternative name: With this extension, identities can be linked to the certificate's subject. These identities can be used in place of, or in addition to, the identity listed in the certificate's subject field.
- Basic constraints: Specifies "*the maximum depth of valid certification paths that include this certificate*" [42], as well as whether the subject of the certificate is a CA.
- Name constraints: All subject names in the following certifications in a certification route must be contained inside the name space indicated by this extension, which can only be used in a CA certificate. The constraint for URIs only applies to the host portion of the name. The constraint may specify a host or domain but must be supplied as a fully qualified domain name. `host.example.com` and `.example.com` are two examples.
- Policy constraints: Certificates deployed to CAs may use the policy constraints extension, which restricts path validation.
- Extended key usage: This extension lists one or more additional uses for the certified public key that can be substituted for or added to those mentioned in the key usage extension. This extension will often only be shown in certificates for end entities.
- Inhibit anyPolicy: Certificates issued to CAs may utilize the inhibit anyPolicy extension. The inhibit anyPolicy extension states that, unless it occurs in an intermediate self-issued CA certificate, the special anyPolicy OID is not considered an explicit match for other certificate policies.

Optimized Format

In the optimized format, entities are not required to receive and parse these extensions, i.e., all extensions are listed as optional. This is performed in order to reduce communication and computational overhead.

3.11.10. TBS Certificate

X.509 Specification

This TBS ("*To Be Signed*") field includes the subject and issuer names, a public key related to the subject, a validity time, and other relevant data [42].

Optimized Format

Because this information is already listed in fields mentioned before (Sections 3.11.4 and 3.11.6), this field will be omitted altogether.

3.11.11. Signature Algorithm

X.509 Specification

The name of the cryptographic algorithm that the CA used to sign this certificate. The value of this field must be the same as in the signature field (Section 3.11.3).

Optimized Format

By restricting the signature algorithm to a single algorithm, the field can be omitted entirely. The signature algorithm chosen is ECDSA NIST P-256.

3.11.12. Signature Value

X.509 Specification

This field contains the tbsCertificate's ASN.1 DER-encoded digital signature.

Optimized Format

Because this field is essential for the proper functioning of the PKI, it is included in the optimized format, without any modifications.

3.11.13. Summary

An overview of the optimized X.509 certificate for IoT devices is shown in Figure 11.

```
TBSCertificate  ::=  SEQUENCE  {
        serialNumber        int
        issuer              string
        validity            UTCTime
        subject             int
        subjectPublicKeyInfo  string
        extensions          list
    }

Certificate  ::=  SEQUENCE  {
        signatureValue      string
    }
```

**Figure 11.** Overview of the optimized X.509 certificate for IoT.

### *3.12. Security Considerations*

We will now present a number of security considerations for the proposed solution, along with security solutions.

#### 3.12.1. Denial-of-Service

Malicious nodes and zones are able to overwhelm the intended victim zone masters/nodes by sending them protocol messages (e.g., certificate lookup, certificate update, and enrollment requests) at a very high frequency. This can result in the victim zone master/node being overwhelmed and shutting down, essentially performing a DoS attack.

To mitigate this issue, all nodes and zone masters will implement rate limiting, which is a mechanism employed to regulate the frequency of requests exchanged by a network interface. If the traffic rate at a node or zone master exceeds a certain threshold, the zone master or node will simply drop the remaining traffic. Although this can have a negative effect on benign nodes, it will prevent DoS attacks.

#### 3.12.2. Blocking of Messages

A malicious zone master is able to block any requests sent between zones. Examples of such requests are zone extension, node enrollment, certificate lookup, and zone master revocation. If a zone master suspects a node to block requests, it will send three distinct dummy requests to the suspicious zone master. When none of the requests receive any response, it will revoke the zone master. If the suspicious zone master is no direct parent or child of the zone master, it will ask neighboring zones to revoke. Moreover, if a node suspects a zone master to be blocking requests, it will tell its zone master who will then perform this (three requests) test on their behalf.

In this section, we have designed a decentralized PKI architecture that employs zones, in which the zone masters act as CAs for all nodes in the master's zone. Furthermore, we have designed a new type of X.509 certificate that will be more lightweight than its traditional variant. It has been designed by removing or reducing the size of the fields present in a classical X.509 certificate. In the next two sections, we will validate our design by conducting a performance (Section 4) and security (Section 5) analysis.

### 4. Performance Analysis

This section presents the design and execution of the framework for evaluating the performance of the proposed solution. The proposed solution has been implemented in Python 3.10, and uses the `pycryptodome` (https://pypi.org/project/pycryptodome/ accessed on 10 December 2023) and `cbor2` (https://pypi.org/project/cbor2/ accessed on 10 December 2023) libraries for cryptographic operations and CBOR encoding, respectively. The implementation is hosted on a Raspberry Pi 4 Model B with 1 GB of LPDDR4-3200 SDRAM memory, and a Broadcom BCM2711, Quad core Cortex-A72 (ARM v8) 64-bit CPU @ 1.8 GHz. The Raspberry Pi employs Alpine Linux (v3.17) as its host OS because it is a lightweight distribution.

Firstly, we will describe the procedure to measure the performance of the implementation. Next, we will present the results of the previously explained methodology.

*4.1. Experimental Setup*

In order to measure the performance of the proposed solution and implementation, we will be performing a time, memory, and power-based analysis.

- In the time analysis, we will first log the current timestamp before executing the operation. After the operation has finished, we will log the timestamp again and compute the difference to obtain the runtime duration.
- In the memory analysis, we will be using `filprofiler` (https://pypi.org/project/filprofiler/, accessed on 10 January 2024), which is a Python tool that analyzes the memory usage of programs.
- In the energy analysis, we will be conducting a baseline voltage and ampere measurement to obtain the baseline usage. When executing the operation, the peak voltage and amperage will be observed, and the differences between the baseline will be noted. By combining the voltage and amperage with the duration of the operation, the energy (J) can be calculated:

$$\text{Energy (J)} = \text{Voltage (V)} \times \text{Amperage (A)} \times \text{Time (t)}.$$

For the measurement, we have used a dedicated Raspberry Pi energy measurement hardware. The experimental setup can be seen in Figure 12.

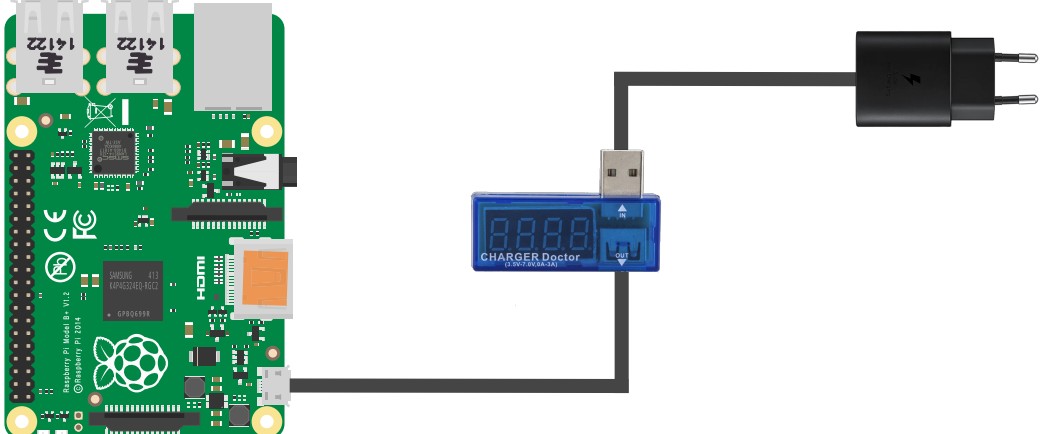

**Figure 12.** Experimental setup for the energy analysis. The dedicated energy measurement hardware sits on the connection between the Raspberry Pi and the power outlet.

In this analysis, we will benchmark the following operations, as we deem these to be most relevant:

- Certificate size
- Certificate generation
- Node enrollment (Section 3.2)
- Zone extension (Section 3.4)
- Certificate lookup (Section 3.7)
- Certificate verification (Section 3.8)

We will benchmark every one of these operations three times. The first run, the benchmark will be performed with traditional X.509 certificates. That is, an X.509 certificate with all fields included. Next, we will run the benchmark with our optimized X.509 certificates with omitted or shortened fields. Finally, we will benchmark our solution with the optimized certificates from the previous run, but we encode them efficiently using CBOR. We want to note that in the first two benchmarks, all certificates are encoded in JSON format for ease of parsing during communication.

We will now describe the four benchmark procedures in more detail.

### 4.2. Certificate Size

The first benchmark will create three certificates, one for each category (traditional, optimized, CBOR encoded). Then, the size of the certificates will be calculated in bytes.

### 4.3. Certificate Generation

In the next benchmark, the time, memory, and energy required to generate a certificate are measured, which includes the signature generation. To this end, we will generate 100 certificates and calculate the average time and memory consumption. As certificates are only CBOR encoded when transmitted across a network, it is not relevant to benchmark this category. Nevertheless, we will test the performance of CBOR encoding to find out whether the generation of such certificates requires fewer resources than regular JSON encoding.

### 4.4. Node Enrollment

This benchmark will create a master and client and make the client connect to the master to enroll. The latter procedure is benchmarked in terms of time, memory, and energy consumption for both the client and master. To this end, the benchmark will be executed 100 times and the average values will be computed.

### 4.5. Zone Extension

This benchmark creates a master and client as well. Next, the client enrolls into the master's zone. Finally, the master instructs the client to extend to a new zone, which is measured in terms of time, memory, and energy consumption.

### 4.6. Certificate Lookup

This benchmark is parametrized by the number of zone masters. That is, this benchmark will first create $x$ zone masters and will enroll the first client in the first zone in the chain and the second client in the final zone in the chain. Finally, the first client will look up the certificate of the second client, whose request is then passed along the entire zone chain. The purpose of this benchmark is mainly to find out whether CBOR-encoded certificates are transferred faster than regular and optimized JSON encoded certificates.

### 4.7. Certificate Verification

The final benchmark will be measuring the memory usage, runtime, and energy usage of certificate verification with five zone masters. Here, a node has to verify the certificate of a node in the final zone and verify the five certificates of the zone masters that make up the certificate chain.

### 4.8. Experimental Results

This section will present the results of the experimental methodology, as described in the previous section. Each subsection will present the results for its respective operation.

### 4.9. Certificate Size

This benchmark generated three certificates, one for each category, and measured their size in bytes. We want to point out that numerous certificates have been generated for every category. Nevertheless, the certificate size remained constant.

As can be seen in Figure 13, the optimization and encoding of the certificates drastically reduces their size. Optimized certificates are 46% smaller than traditional X.509 certificates, and CBOR-encoded certificates are 52% smaller than traditional certificates.

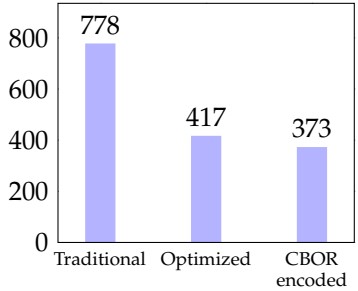

**Figure 13.** Certificate size in bytes.

### 4.10. Certificate Generation

In this benchmark, we generated 100 certificates and calculated the average execution time, memory usage, and energy consumption for traditional, optimized, and CBOR-encoded certificates. Although it can be seen in Figure 14 that optimized certificates take less time and memory to be generated, it is not by a significant margin. Furthermore, it can be noted that CBOR encoding requires less energy to generate than regular JSON encoding. The greatest improvements can be seen in terms of memory, where the CBOR encoding requires > 2000 fewer bytes of memory than when generating a traditional certificate.

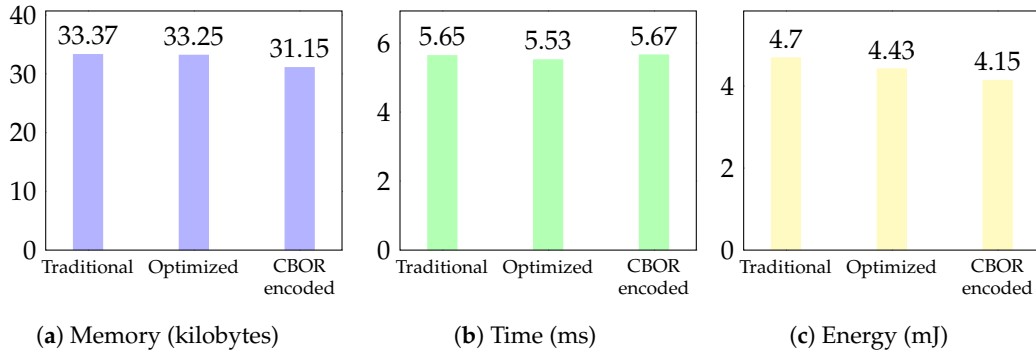

(**a**) Memory (kilobytes)  (**b**) Time (ms)  (**c**) Energy (mJ)

**Figure 14.** Benchmark of certificate generation.

### 4.11. Node Enrollment

In this benchmark, the node enrollment procedure was executed 100 times, and the average duration, energy, and memory consumption of enrollment were calculated. This benchmark was conducted from both the client and master perspective, in order to give a more comprehensive overview.

In Figure 15, it can be seen that the optimized certificate slightly outperforms the traditional certificate, in terms of memory and energy usage. Although the CBOR-encoded certificate seems to perform better than both the traditional and optimized certificate on the client side in terms of time, this is not the case for the master side. Furthermore, it is interesting to note that the master side needs significantly more memory, which is mainly due to the cryptographic operations the master has to perform. Nevertheless, the client side requires more time to complete than the master side, which could be due to the fact that the client has to find a zone master with vacant spots, which could require the client to send numerous requests across the network. Finally, we note that on the client side, both CBOR encoding and optimized certificates do have a positive effect on energy usage, where energy usage is reduced by 33% and 50% on the client and master side, respectively.

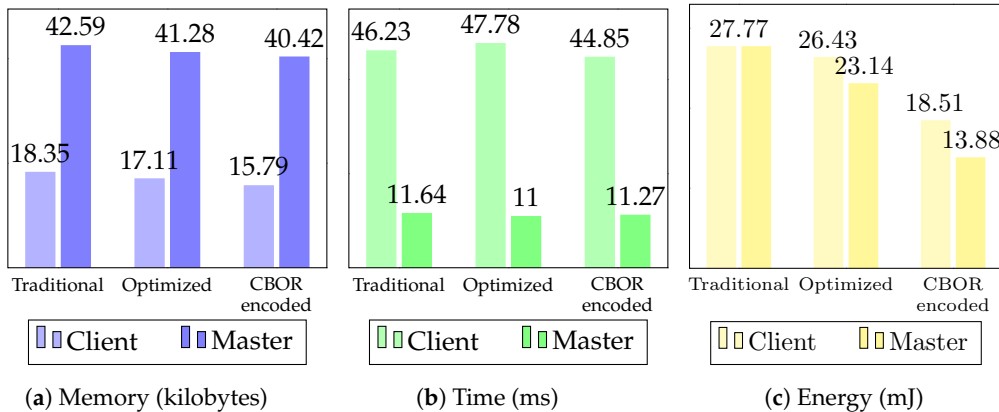

(**a**) Memory (kilobytes)     (**b**) Time (ms)     (**c**) Energy (mJ)

**Figure 15.** Benchmark of node enrollment.

### 4.12. Zone Extension

In this benchmark, the zone extension procedure was executed 100 times, and the average time, energy, and memory consumption during extension was calculated.

It is important to note that the zone extension procedure does not make use of certificates that are sent across the network. Consequently, the efficiency of CBOR encoding is not leveraged as much as with, for example, the node enrollment procedure. Therefore, in this benchmark, we will only compare regular JSON encoding with CBOR encoding.

Figure 16 demonstrates that this benchmark does not show a significant improvement in using CBOR encoding over JSON encoding due to the aforementioned reason.

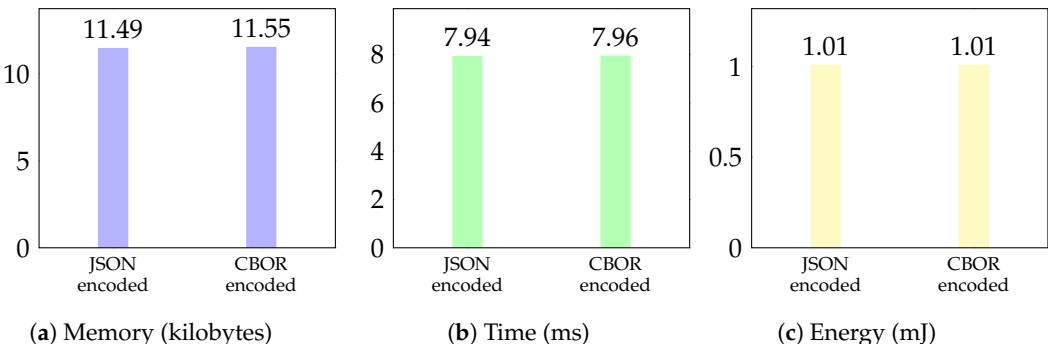

(**a**) Memory (kilobytes)     (**b**) Time (ms)     (**c**) Energy (mJ)

**Figure 16.** Benchmark of zone extension.

### 4.13. Certificate Lookup

In this benchmark, there is one node in the first zone master in the chain and another node in the final zone master. The first node will look up the certificate of the second node so that the zone masters will have to propagate this request along all zone masters in the chain. For this parametrized benchmark, we executed the benchmark for $x$ zone masters, where $x$ is an integer in $[2, 150]$. We chose the upper bound of zone masters to be 150, as the Raspberry Pi was not able to handle more.

As depicted in Figure 17, a noticeable enhancement in execution time can be observed when comparing traditional certificates with optimized and encoded certificates, particularly for zone counts exceeding approximately 20. This is attributed to the fact that an increased number of messages are required to facilitate certificate lookups as the number of zones grows, thus leading to a greater utilization of the CBOR encoding technique. For example, for $x = 150$ zones, the CBOR-encoded certificate performs 12.5% faster than traditional certificates and 10.9% faster than optimized certificates.

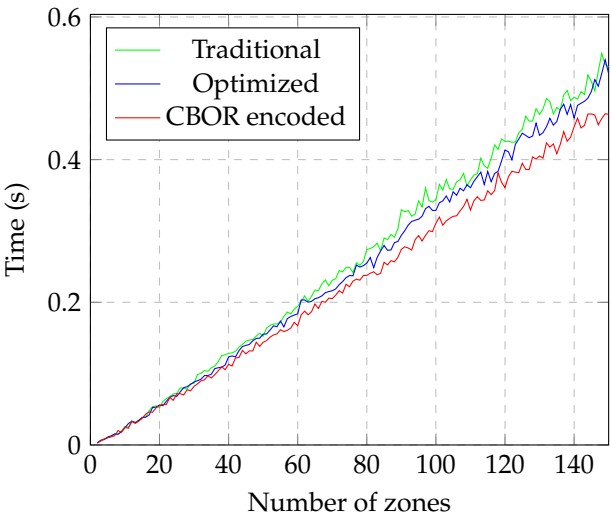

**Figure 17.** Benchmark of certificate lookup.

### 4.14. Certificate Verification

In this benchmark, we will be measuring the memory usage, runtime, and energy usage of certificate verification with five zone masters. Here, a node has to verify the certificate of a node in the final zone and verify the five certificates of the zone masters that make up the certificate chain.

As can be seen in Figure 18, while in terms of memory, the CBOR-encoded certificate does not yield any improvements over the traditional certificate, there are minor improvements in runtime. Furthermore, we can observe that there are significant energy usage improvements for optimized and CBOR-encoded certificates.

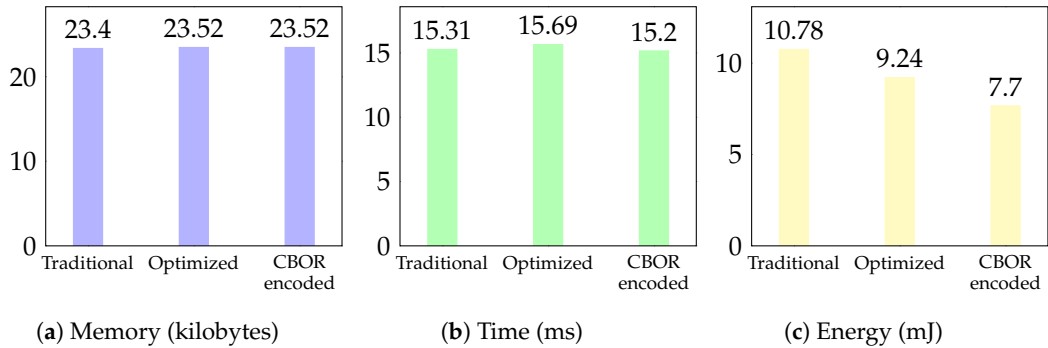

(**a**) Memory (kilobytes)　　　　(**b**) Time (ms)　　　　(**c**) Energy (mJ)

**Figure 18.** Benchmark of certificate verification.

### 4.15. Discussion

This section will discuss the obtained performance results by comparing them to results found in the literature.

From the tables that have been created during the literature review, two types of metrics have been found that allow for comparison between works: certificate size and certificate verification, in terms of runtime and energy usage.

### 4.16. Certificate Size

For certificate size, we have the work by Forsby et al. [34], where the authors suggest a lightweight X.509 certificate format made especially for Internet of Things devices. This novel format seeks to provide safe authentication and data integrity while reducing certificate size and computational overhead. The lightweight certificate shows compliance with the X.509 standard, which allows use in all current PKI implementations. Not only is our optimized certificate larger than that of [34] (417 vs. 324 bytes), this is also the case

for the compressed certificate (373 vs. 146 bytes). This can be due to the fact that [34] has applied more efficient techniques for optimizing certificate size. For example, for the "Subject Public Key Info" field, [34] have taken extra steps to compress the ECC keys, even before CBOR compression. Furthermore, another reason for smaller certificate sizes could be attributed to the fact that the CBOR libraries used in both works are implemented differently. Namely, the library in our work (cbor2) is noted to be compliant with the original CBOR specification (RFC 7049, [44]). Nevertheless, the CBOR library used in [34] has been implemented before the existence of RFC 7049 [44].

*4.17. Certificate Verification*

For certificate verification time and energy consumption, we have the works by Won et al. [4], Singla and Bertino [26], Pinol et al. [39], and Marino et al. [21]. Firstly, it can be seen that our work outperforms [4] and [26] by 8.4–16.4 times. Nevertheless, due to the fact that the architectures in these works are blockchain-based, and thus completely different from our work, we will not go into much detail as we do not think that would be a fair comparison.

Next, we have Pinol et al. [39], who created an open-source ECC that is optimized for the IoT and includes an Elliptic-Curve Diffie-Hellman (ECDH) and an Elliptic-Curve Digital Signature Algorithm (ECDSA). They skip division operations since they are computationally expensive in order to optimize the algorithms. They instead use modular arithmetic. The comparison shows that our implementation is 263–724 times faster than that of [39], depending on the key size. This is mainly due to the fact that the implementation of [39] has been tested on an STM32W108CC wireless chip, which has much slower hardware than a Raspberry Pi 4B [45]. Furthermore, another reason could be that the authors have implemented their own ECC, instead of relying on libraries. This could introduce performance bottlenecks, as custom ECC implementations could be non-compliant with specifications.

The final work up for comparison is that by Marino et al. [21], who developed PKIoT, which allows IoT nodes to assign computationally intensive security-related tasks to a remote server. Nodes are free to decide which tasks to assign based on their current state and level of trust in the server. As a result, the PKIoT architecture offers a scalable, adaptable, and flexible solution. They also developed a novel type of compact certificate, which requires PKIoT compatibility on both ends of the communication but allows for even further transmission overhead reductions when used in place of normal X.509 certificates. The comparison results show that our implementation is almost 31 times faster, and consumes two times less energy than the work in [21]. This can be attributed to a number of factors. Firstly, the usage of different crypto libraries: while our implementation uses pycryptodome [21], reference [46]uses micro-ecc . Due to implementation and programming language differences, these libraries can achieve different results in terms of performance. Nevertheless, the improvement in performance cannot be attributed to our hardware; the PKIoT server, used to perform cryptographic operations when nodes choose so, is a PC with an Intel Core i5-6500 @ 3.20 GHz and 15.6 GB of RAM. However, it is possible that the worse performance can be due to certificate size. In [21], on the client side, certificates are stored as a simple link to the full certificate, which is located on a PKIoT server. This certificate is unfortunately a full-sized certificate, which can explain the longer verification times and higher energy consumption.

We conclude this section by giving a comprehensive overview of the discussed articles in Table 2.

**Table 2.** Comparison of proposed solution with those present in the literature.

| Type of Metric | Name | Authors' Results | Our Results | Improvement |
|---|---|---|---|---|
| Certificate size | Forsby et al. [34] | ● Optimized:  324 bytes<br>● Compressed:  146 bytes | ● Optimized:  417 bytes<br>● Compressed:  373 bytes | ✗ |
| Certificate verification | IoT-PKI [4]<br>Pinol et al. [39]<br>Singla et al. [26]<br>PKIoT [21] | ● 128 ms<br>● 4–11 s, 153.84 mJ<br>● 128–250 ms<br>● 469 ms, 16.57 mJ | ● 15.2 ms<br>● 7.7 mJ | ✓<br>✓<br>✓<br>✓ |

*4.18. Conclusions*

In benchmarks, Sections 4.10–4.12, it is shown that there is no significant difference in execution time between the various types of certificates. Nevertheless, the benchmarks do mostly show that the optimized CBOR-encoded certificates perform better in terms of memory usage, certificate size, and energy consumption. This can be attributed to a number of factors:

1.  More effective number encoding is supported by CBOR, which employs a variable-length integer encoding system that consumes fewer bytes than JSON to represent small values. In contrast to JSON's text-based representation, CBOR permits the encoding of floating-point integers using a more compact binary format [44].
2.  CBOR uses a length-prefix encoding strategy for Unicode strings, which enables it to represent strings using fewer bytes than JSON's Unicode escape sequences. This enables CBOR to support more compact encoding of Unicode strings. For instance, the Unicode characters in CBOR are never escaped, in contrast to formats like JSON. Therefore, a newline character (U+000A) is never represented in a string as bytes `0x5c6e` (the characters "\" and "n") but rather as bytes `0x0a`.
3.  Tagging is one of the characteristics of CBOR, which may enable it to represent some categories of data more effectively than JSON. For instance, CBOR comes with built-in support for expressing dates and times, which can be achieved with less data than equivalent JSON expressions.

What is surprising is that the CBOR encoding seems to have no significant effect on the performance in terms of time. In fact, although the differences are barely noticeable, most benchmarks perform slower than their optimized JSON counterparts. We argue that CBOR requires more time to encode/decode than JSON, as CBOR needs to compress/decompress the data as well, and because of the aforementioned reasons.

Furthermore, we argue that optimized and CBOR-encoded certificates have a positive effect on energy usage when compared with the traditional certificates. This can be attributed to the fact that CBOR encoding requires less resources than regular JSON encoding. It is important to note that the energy measurement results could be skewed due to hardware inaccuracies.

Also, certificate lookup is visibly improved using CBOR as long as the number of zones is high enough. We suppose CBOR encoding only has an effect on operations that require CBOR data to be sent over the network, such as certificate lookup, therefore leveraging the power of efficient encoding the most.

We conclude that CBOR encoding has an insignificant or adverse effect on all operations in terms of time because the compression of CBOR encoding requires more time than regular JSON. Nevertheless, when CBOR-encoded certificates are sent across numerous hosts (Section 4.13). In such cases, the power of CBOR encoding is properly leveraged. Furthermore, the benchmarks mostly show that the CBOR encoding shows a significant decrease in memory consumption and certificate size. By comparing our obtained results with those from the literature, we have concluded that our certificate verification is faster and consumes less energy, but it must be taken into account that different architectures and hardware platforms could have had an effect on these outcomes. Nonetheless, we also

concluded that there is still room for improvement in certificate size, as other works have shown that a smaller certificate size can be obtained.

The next section will validate our proposed solution in terms of security, by conducting a theoretical and formal security analysis.

## 5. Security Analysis

This section presents details of the security analysis that will be performed on the implementation. The security analysis is three-fold. Firstly, we will be conducting a theoretical security analysis using theoretical proofs. Secondly, we will be outlining a number of security guarantees, such as resilience against replay attacks. Finally, we will be performing a formal analysis using a formal verification tool called Automated Validation of Internet Security Protocols and Applications (AVISPA). However, first, we will describe the adversarial model and assumptions.

### 5.1. Adversarial Model

An adversary will possess the following capabilities:

- The adversary will have unrestricted control over the communication channel, enabling them to observe, manipulate, or replay the messages conveyed over the channel.
- The adversary has the capability to employ the following attack model and execute it within polynomial time:
    - `lookup(`$N$`, `$M$`)` This operation allows the adversary to legitimately request the public key of node $N$ through zone master $M$, as if it were a regular user.
    - `enroll(`$M$`)` This operation allows the adversary to legitimately enroll into a zone with master $M$, as if it were a regular user.
    - `update(`$C$`, `$M$`)` This operation allows the adversary to update its certificate to $C$ using zone master $M$.
    - `revoke(`$Z$`)` This operation allows a malicious zone master to revoke its child or parent zone $Z$.
    - `corruptNode()` This attack model represents a scenario in which a node becomes compromised or corrupted.
    - `corruptZoneMaster()` This attack model represents a scenario in which a zone master becomes compromised or corrupted.

### 5.2. Assumptions

- The adversary:
    - Can start any number of parallel protocol sessions,
    - Knows the functioning of the entire protocol,
    - Can build and send messages,
    - Can read, retain, and block any sent message,
    - Can decrypt any message for which it has the key.
- It is computationally infeasible for the adversary to break the Elliptic-Curve Digital Signature Algorithm (ECDSA), Elliptic-Curve Diffie-Hellman (ECDH), and Elliptic-Curve Discrete Logarithm Problem (ECDLP).

### 5.3. Theoretical Analysis

The architecture employs Elliptic Curve Cryptography (ECC) to generate its keys and signatures, as outlined in Section 3.11. Additionally, the SHA-256 hashing algorithm is utilized as the secure hashing algorithm. This section will provide a theoretical security analysis of various operations present in the design of the PKI.

### 5.3.1. Certificate Updating

In the case that a node wants to update its public-private keypair, it can issue a certificate update request to the zone master. It must be ensured that only the node that is in possession of the secret key is able to update its keypair.

**Theorem 1.** *A node who does not have the secret key for an identity is unable to send a legitimate update transaction to the zone master.*

**Proof.** If a node wants to update its keypair, it can simply send a new certificate $C_{new}$ signed with the old key $SK_{old}$. In order to create this signature $S$, the node needs to be in possession of $SK_{old}$. The submission of an update transaction by an adversary, for an identity that is not under their possession, can only be achieved through the successful reconstruction of $SK_{old}$ and the creation of a valid signature for the new certificate. If the adversary is able to do this, it would imply that the adversary is able to break the ECDSA, which is computationally infeasible for a computationally limited adversary. □

### 5.3.2. Node Unenrollment

When a node wants to unenroll from the zone, it can request the zone master to do so, who will then revoke the certificate of the node in its internal table. The node does this by sending a signed message requesting to unenroll, similar to the certificate update procedure.

**Theorem 2.** *It is impossible for an adversary to submit a legitimate revoke transaction to the zone master for an identity that they do not hold the secret key for.*

**Proof.** If a node wants to unenroll from the zone, it will send a request signed with $SK$ to the zone master. If an adversary wants unenroll a node for which it does not own the private key $SK$, it will need to obtain a hold of $SK$. Therefore, the adversary must be able to break the ECDSA, which is computationally infeasible for a computationally bound adversary. □

### 5.3.3. Certificate Lookup

When two nodes want to communicate securely, they will have to look up their respective certificates. This can be achieved by querying the zone master for the certificates. It is, therefore, essential that the certificates are not modified in transit. We propose the following theorem:

**Theorem 3.** *It is impossible for an adversary to modify a certificate in transit during a lookup procedure.*

Every certificate handed out by a zone master is signed by the zone master itself. Furthermore, every zone master public key certificate is signed by its parent. Therefore, if a node wants to check the validity of a certificate, it can check the signature on the certificate. Next, it will traverse the certificate chain until it reaches the root certificate, for which it must assume that it is to be trusted. Therefore, if an adversary aims to perform an MITM with a malicious certificate, it must be able to forge the signature of the zone master in which a node resides. Thus, the adversary must be able to obtain the private key of the zone master or must be able to break the ECDSA, which is computationally infeasible given a computationally limited adversary.

### 5.4. Security Guarantees

This section will outline a number of security guarantees for the proposed PKI solution, but to begin with, we will outline the security goals/requirements within a theoretical adversary model, derive theorems with proofs, and then compare them with existing traditional PKI systems.

5.4.1. Theoretical Adversary Model

**Adversarial Capabilities:**

- The adversary has unrestricted control over the communication channel, enabling them to intercept, manipulate, or replay messages.
- The adversary possesses the capability to execute attacks within polynomial time.

**Assumptions:**

- The cryptographic algorithms used in the PKI system, such as ECDSA for digital signatures and AES for encryption, are secure and resistant to known attacks.
- The adversary cannot break the underlying cryptographic algorithms, such as ECDSA, AES, and SHA-256, within a reasonable computational time frame.

5.4.2. Security Goals/Requirements

1. Authentication
   **Theorem 1:**

   (a) *Statement:* In the proposed lightweight PKI system, every message sender can be authenticated using digital signatures.
   (b) *Proof:* Let $M$ be a message sender who wants to authenticate themselves. By verifying the digital signature attached to $M$'s message using the sender's public key, the recipient can confirm the sender's identity.

2. Data Integrity
   **Theorem 2:**

   (a) *Statement:* Data exchanged between entities in the lightweight PKI system remains unchanged during transmission.
   (b) *Proof:* By employing cryptographic hash functions (e.g., SHA-256), the sender can compute a hash of the message before transmission. Upon receipt, the recipient can independently compute the hash and compare it with the received hash to verify data integrity.

3. Confidentiality
   **Theorem 3:**

   (a) *Statement:* Messages exchanged between entities are encrypted and cannot be deciphered by unauthorized parties.
   (b) *Proof:* Encryption algorithms such as AES are employed to encrypt messages before transmission. Without access to the encryption key, unauthorized parties cannot decipher the encrypted messages.

5.4.3. Comparison with Traditional PKI

**Authentication:** In traditional PKI systems, authentication relies on digital certificates issued by trusted certificate authorities (CAs). However, the lightweight PKI system achieves authentication through decentralized mechanisms without the need for centralized CAs, reducing dependency and potential single points of failure. Data Integrity: Both traditional and lightweight PKI systems utilize cryptographic hash functions to ensure data integrity. However, the lightweight PKI system may offer more efficient and lightweight implementations suitable for resource-constrained IoT devices.

**Confidentiality:** Both systems employ encryption algorithms to achieve confidentiality. However, the lightweight PKI system may prioritize lightweight cryptographic techniques, such as ECC, to optimize performance without compromising security, making it more suitable for IoT environments.

**Overall Security:** While traditional PKI systems may offer a more established and widely recognized framework, the lightweight PKI system addresses the specific security challenges of IoT environments, such as resource constraints and scalability, through innovative approaches such as decentralized architecture and lightweight cryptography.

### 5.4.4. Secure against Replay Attacks

Our PKI is resilient against replay attacks because the authenticity of messages sent over public key cryptography can be verified by checking the certificate, along with the signatures of all zone masters. Furthermore, messages sent over public key cryptography include encrypted nonces, which ensure that an adversary is not able to retransmit such messages (Section 3.3). The only potential possibility for replay attacks is because, for node enrollment, an adversary can simply retransmit an enrollment request. Nevertheless, this does not have a major impact, as the zone master can simply check that a certain node has already been enrolled. If an adversary stores an enrollment request for a node that has unenrolled out of the zone, replay attacks are prevented by requiring a fresh Diffie–Hellman key exchange (that is, with new keys) for every enrollment request, regardless of whether this node has enrolled before.

### 5.4.5. Mutual Authentication

Our system offers mutual authentication, in which all entities (such as IoT nodes and zone masters) mutually verify one another's identities. More specifically, because it is computationally infeasible to break the ECDH, adversaries are unable to spoof authentication on enrollment requests $\lambda = \{([PK], H)\}_\sigma$, as they are encrypted with a symmetric key $\sigma$. Furthermore, adversaries cannot spoof messages exchanged over public key cryptography as they are protected by an encrypted nonce (Section 3.3).

### 5.4.6. Secure against Impersonation Attacks

An attacker is unable to create legitimate messages to send over the network unless it has access to the zone master's or an IoT node's private key. Therefore, it is impossible for an adversary to pretend to be a zone master or an IoT node.

### 5.4.7. Secure against Man-in-the-Middle Attacks

Man-in-the-middle attack (MITM) attacks occur when an active attacker successfully poses as both the user to the server and the server to the user by intercepting the communication connection between a legitimate user and the server. Afterward, both the user and the targeted server will think they are speaking to one another. From the previously mentioned guarantees, we infer that our protocol can offer mutual authentication, which enables us to prevent MITM attacks.

### 5.4.8. Secure against Brute-Force Attacks

The total number of unique keys used in the encryption system is what determines the size of the key space. In order to prevent brute-force attacks, an encryption algorithm's key-space needs to be sufficiently vast. Our proposed PKI employs the NIST P-256 curve, which has a key size of 256 bits. Therefore, the entire keyspace of this encryption scheme is $1.16 \times 10^{77}$. Following, we conclude that a brute force attack is computationally infeasible for such a keyspace.

### 5.4.9. Secure against Passive Attacks

To prevent adversaries from listening in, the transmitted data are encrypted in the proposed solution using the ECC and AES encryption scheme. As a result, without the decryption key, the passive adversaries are unable to decrypt the intercepted message. Thus, we conclude that our proposed solution is resilient against passive attacks.

### 5.5. Formal Analysis

For the formal analysis, we have employed the Automated Validation of Internet Security Protocols and Applications (AVISPA) tool, which is frequently used to examine security protocols and cryptographic properties. Researchers and developers can use it as a platform for automatic assessment and testing of security aspects of protocols [47]. AVISPA examines the provided model under the presumptions of perfect cryptography [48]

and protocol message exchange via a network controlled by a Dolev-Yao adversary [49]. Specifically, the intruder can intercept messages and analyze them if it has the decryption keys, and it can act upon this knowledge.

In this analysis, two models have been written in High-Level Protocol Specification Language (HLPSL), which AVISPA internally translates to an Intermediary Format (IF), which it feeds to a backend, Constraint-Logic-based Attack Searcher (CL-Atse). We opted to write two HLPSL models, as we deemed these models to cover all operations inside the proposed design best:

1. Messages over symmetric-key cryptography: in this HLPSL specification, the node enrollment procedure (Section 3.2) is modeled. That is, the node and zone master agree on a symmetric key using a Diffie–Hellman key exchange. Next, the node sends a symmetrically encrypted request to the zone master to be enrolled in the zone. Finally, the zone master responds to the enrollment request with a symmetrically encrypted message as well.

2. Messages over public-key cryptography (Section 3.3): in this specification, all other operations in the design are modeled. Firstly, the requesting party generates a nonce and sends this nonce, along with the request payload in an encrypted form (with the public key of the receiving party) to the receiver. The receiving party responds with the same nonce, concatenated with its desired response payload. The entire response is encrypted with the public key of the sending party. The reason for including a nonce in the request and response is to prevent MITM attacks; more details are presented in Section 3.3.

The full HLPSL models can be found in Appendix A.

Results

Both proposed models have been tested using the CL-Atse backend. The simulation results showed that both models protect the confidentiality of the message payloads, and that the second HLPSL model ensures correct authentication of both communicating parties. We want to note that checking for this property in the first model is not relevant, as it is assumed that all entities with possession of symmetric keys are authenticated parties, while in the second model, any unauthenticated party can send messages encrypted with the public key of the recipient. According to the best of our knowledge, both HLPSL specifications cover the entire protocol design in a complete way. Therefore, we consider, according to the formal analysis, the entire design to be secure.

In this section, we have performed a theoretical and formal analysis of the proposed PKI design. Furthermore, we have provided a number of security guarantees, such as resilience against impersonation attacks. In the theoretical analysis, we have written a number of theorems related to various operations inside this new architecture, along with their proofs. Next, in the formal analysis, we have written to HLPSL specifications that model two types of communication in the architecture: (i) those using symmetric-key cryptography, and (ii) those using public-key cryptography. Assuming that the two specifications fully cover the entire architecture, we conclude that along with the theoretical analysis and the security guarantees, the proposed solution is secure.

## 6. Conclusions

In this article, we have researched the academic landscape regarding Public Key Infrastructure (PKI) for the Internet of Things (IoT) in order to identify issues and opportunities in designing such a PKI. Furthermore, we have designed, implemented, and evaluated a lightweight PKI tailored to IoT devices. This section aims to conclude this article, by noting the research objectives, answering the research questions, stating the contributions, limitations, and potential future works.

### 6.1. Research Objectives

The term Internet of Things (IoT) describes actual physical items with sensors, processing power, and software that may connect to other systems and devices via the Internet to exchange data [1]. The IoT is converting people's immediate surroundings into a cyberphysical system that they can interact with by using wearables and personal mobile devices like smartphones. However, there are many security issues in the IoT space that need to be resolved [6]. Researchers have documented numerous attempts to compromise the availability [13,14], confidentiality [7–9], and integrity [10–12] of IoT devices. Public Key Infrastructure (PKI) is one of many solutions to address the aforementioned issues and is already used daily in traditional computer systems. PKI facilitates the generation, dissemination, modification, and revocation of digital certificates. A digital certificate serves as a connection between public keys and the identities of numerous entities [50]. As part of the procedure to create the connection, a Certificate Authority (CA) produces and registers these digital certificates.

Unfortunately, IoT vendors have been slow to deploy PKI for a number of reasons:

1. The CA introduces a single point of failure due to the design of conventional PKI infrastructures. Therefore, if an attacker manages to take over the CA, they will be able to issue certificates for whomever they choose. This problem is particularly crucial in IoT contexts because the vast array of connected devices increases the possible attack surface.
2. Traditional cryptographic algorithms will have severe performance constraints when implemented on regular IoT hardware due to the resource-constrained nature of IoT devices. To illustrate, Blanc et al. [18] have ported 12 encryption algorithms to multiple IoT hardware platforms to compare their performance. Their experimental results show that the algorithms run 6–50 times slower than on traditional PCs, depending on the architecture.

By creating a decentralized PKI system that uses lightweight cryptography and is suitable for IoT devices with minimal processing power, this paper seeks to overcome the aforementioned problems. The aforementioned issues emphasize the importance of this research, as do the numerous attacks on IoT devices that have been documented [7–15,19].

### 6.2. Contributions

In this article, we have provided two contributions. Firstly, an SLR on the academic landscape regarding PKI for the Internet of Things (IoT). This SLR has resulted in more than 30 articles, which have been analyzed based on metadata and content. From this analysis, we have concluded that in the current academic landscape, PKI solutions make use of mostly five technologies: Elliptic Curve Cryptography (ECC), decentralized technologies, DNA cryptography, pairing-based cryptography, and Physical Unclonable Functions (PUFs). We have summarized all 37 articles and provided a concise overview of all advantages and disadvantages of the five technologies found in the SLR. We end the SLR by giving a comprehensive overview of all articles by listing their features, such as technology used, and the results of their performance and security analyses. To the best of our knowledge, no such SLR has been conducted before, thus stressing the importance of this contribution.

The second contribution of this paper is the design, implementation and evaluation of a Public Key Infrastructure (PKI), tailored to Internet of Things (IoT) devices. This implementation aims to address the research objectives stated earlier, by creating a PKI that is not only lightweight in terms of computational requirements but that also scales well in terms of IoT devices. These two aspects are especially important in the IoT context since such devices are characterized by having stringent resource constraints and are expected to come by the billions in the upcoming years [19]. We have not only found that it is feasible to implement a PKI for IoT, but that this PKI, in some aspects, performs better than existing PKIs in the literature (Section 4.15). This stresses the importance of this

contribution, as secure communication is very important among IoT devices, and should not introduce especially significant computational and communication overhead.

*6.3. Research Questions*

In this section, we aim to answer the research question proposed at the start of this article (Section 1).

6.3.1. RQ1: How Do We Implement a Lightweight PKI Solution for the IoT?

As the IoT environment expands quickly, connecting everything from sensors to appliances, there is a growing need for a scalable and decentralized PKI. The specific security concerns that IoT ecosystems raise can be addressed using a decentralized, lightweight PKI. First, given the nature of IoT networks, a PKI architecture for the IoT must be decentralized. Decentralized PKIs increase attack resilience by doing away with the requirement for a centralized authority to manage certificates, which reduces single points of failure. Second, a lightweight PKI system is a PKI system designed specifically for IoT devices with constrained resources. These devices usually have little memory, computation, and energy capacities. Traditional PKI architectures, which were initially developed for more powerful computing environments, can hinder the performance and effectiveness of IoT devices.

We have addressed these two points by implementing a lightweight PKI. The contributions of this lightweight PKI are two-fold:

1. We have proposed a decentralized PKI system as a solution for the single point of failure that traditional PKI architectures have. We have suggested introducing an architecture that makes use of "zones", each of which has a master. All other IoT nodes in its zone have a master that issues, updates, revokes, and searches for certificates for them. Consequently, compared to the conventional PKI design, it functions much like a Certificate Authority (CA). Zone masters are responsible for keeping track of all nodes' certificates within their zone. If a zone has more nodes than it can manage, one of the nodes in the zone will be changed to a new master node, forming a new zone that is connected to the old zone. A parent–child architecture is created as a result, with links between each zone (aside from the root zone).

2. The second contribution has addressed the need for a lightweight architecture, which is achieved in two following ways: (i) Elliptic Curve Cryptography (ECC), a public key cryptography based on the algebraic structure of elliptic curves over finite fields, is a technique for using lightweight encryption. In comparison to traditional cryptographic techniques like RSA, it has an advantage because it provides the same level of security with smaller key sizes [30]. (ii) The usage of lightweight certificates: in order to speed up certificate creation and verification, the suggested design will drastically reduce the amount and size of fields included in the current X.509 certificates. We have encoded the certificates using Concise Binary Object Representation (CBOR) to further reduce communication overhead.

6.3.2. RQ2: How Will This New Lightweight PKI Solution Perform Compared with Traditional and Literature PKI Solutions?

We have gathered five articles from the systematic literature of [20] that are eligible for a performance comparison. These five articles have performance results across two types of metrics: certificate size and certificate verification (in terms of energy consumption and time).

Only one article has discussed the first type of metric, from the comparison between the article and our work, we have found that there is still room for improvement: our optimized and CBOR-encoded certificates are 28.7% and 155.47% larger, respectively, than those in the work. We have also argued why this could be the case.

The other four articles have discussed the other metric—certificate verification. From this perspective, our implementation performs much faster and more efficiently than the works in the literature. Namely, our implementation is 8.4–724 times faster and 2.1–20 times more

energy efficient, when looking at all works. Furthermore, we have argued why our implementation is more efficient. Nevertheless, it must be noted that differences in architectures and hardware performance can have a large influence on such performance results.

*6.4. Limitations and Future Works*

This section describes some limitations of the proposed PKI solution, along with potential future works to address the limitations.

### 6.4.1. Certificate Queries

Nodes are not updated on the new certificates of other nodes due to the architecture's design. To share data securely, nodes must periodically ask the zone master for certificates. The zone master may be subject to a heavy workload as a result. This is a problem that must be solved with, for example, client-side certificate caching. Nodes can cache certificates so that they do not have to query the zone master every time they want to communicate.

### 6.4.2. TOFU on Enrollment

The node enrollment strategy used by the current system is based on the idea of Trust on First Use (TOFU) [33]. Essentially, the zone master assumes that the node is benign during this process and will not falsify its identity, as indicated by its UUID, when generating its certificate. For future work, this issue can be solved, potentially by a cryptographic verification mechanism that introduces unique fingerprints for each IoT node so that zone masters can distinguish between nodes.

### 6.4.3. Implementation Bottleneck

The proposed solution has been implemented in Python, with the reason that it is only a proof-of-concept. While the implementation can be deployed on hardware such as a Raspberry Pi, this is not the case for smaller hardware such as an ESP32. For a potential future work, the proposed solution can be implemented in a lightweight variant of Python such as MicroPython (https://micropython.org/, accessed on 25 December 2023), or an even lower-level language such as C.

### 6.4.4. Certificate Verification

If a node wants to verify a certificate of a node, it has to traverse the entire certificate chain from the node's zone master, all the way to the root zone. To this end, it has to request the certificates and signatures of all mentioned zone masters. This can impose a significant burden on the first number of zone masters, potentially bottlenecking the architecture. A potential solution could be to verify only several parent zone masters, while still ensuring secure certificate verification.

In conclusion, our proposed solution addresses the critical shortcomings of traditional PKI architectures by introducing a decentralized system designed to mitigate the risk of single points of failure. Through the implementation of "zones" overseen by zone masters, our architecture ensures efficient certificate management and distribution within distinct IoT clusters. This hierarchical structure allows for scalability and resilience, with the flexibility to adapt to dynamic changes in node density. Furthermore, our solution prioritizes lightweight architecture to accommodate the resource constraints inherent in IoT environments. By leveraging Elliptic Curve Cryptography (ECC) and optimizing certificate structures, we significantly enhance efficiency without compromising security. The adoption of lightweight certificates, coupled with the utilization of Concise Binary Object Representation (CBOR) for encoding, minimizes communication overhead and streamlines certificate operations. Through these innovations, we present a robust and scalable PKI framework tailored to the unique requirements of IoT deployments, reinforcing the importance of addressing security and efficiency challenges in the rapidly evolving IoT landscape.

**Author Contributions:** Conceptualization, M.E.-H. and P.B.; methodology, P.B.; software, M.E.-H. and P.B.; validation, M.E.-H. and P.B.; formal analysis, M.E.-H. and P.B.; investigation, M.E.-H.; resources, M.E.-H. and P.B.; data curation, M.E.-H. and P.B.; writing—original draft preparation, P.B.; writing—review and editing, M.E.-H.; visualization, P.B.; supervision, M.E.-H.; project administration, M.E.-H. All authors have read and agreed to the published version of the manuscript.

**Funding:** This research received no external funding.

**Institutional Review Board Statement:** Not applicable.

**Informed Consent Statement:** Not applicable.

**Data Availability Statement:** Data are contained within the article.

**Conflicts of Interest:** The authors declare no conflict of interest.

## Appendix A. HLPSL Models

*Appendix A.1. Node Enrollment (Section 3.2)*

**Listing A1.** Node Enrollment.

```
1  role role_N(N:agent,S:agent,K:symmetric_key,SND,RCV:channel(dy))
2    played_by N
3    def=
4    local
5    State:nat,
6    R:text,C:text
7    init
8    State := 0
9    transition
10   1. State=0 /\ RCV(start) =|>
11   State':=1
12   /\ R':=new()
13   /\ SND({R'}_K)
14   2. State=1 /\ RCV({C'}_K) =|>
15   State':=2
16   end role
17
18  role role_S(S:agent,N:agent,C:text,K:symmetric_key,SND,RCV:channel(dy))
19    played_by S
20    def=
21    local
22    State:nat,
23    Nonce:text,R:text
24    init
25    State := 0
26    transition
27    1. State=0 /\ RCV({R'}_K) =|>
28    State':=1
29    /\ SND({C}_K)
30    /\ secret(C,sec_1,{N,S})
31    end role
32
33  role session(N:agent,S:agent,C:text,K:symmetric_key)
34    def=
35    local
36    SND1,RCV1,SND2,RCV2:channel(dy)
37    composition
38    role_S(S,N,C,K,SND1,RCV1) /\ role_N(N,S,K,SND2,RCV2)
39    end role
40
41  role environment()
42    def=
43    const
44    k:symmetric_key,
45    node,server:agent,
46    c:text,
47    sec_1:protocol_id
48    intruder_knowledge = {node,server}
```

```
49    composition
50    session(node,server,c,k)
51    end role
52
53 goal
54    secrecy_of sec_1
55 end goal
56
57 environment().
```

*Appendix A.2. Subsequent Communication (Section 3.3)*

**Listing A2.** Subsequent Communication.

```
1  role role_N(N:agent,S:agent,Kn:public_key,Ks:public_key,SND,RCV:channel(dy))
2    played_by N
3    def=
4    local
5    State:nat,Nonce:text,Request,Response:text
6    init
7    State := 0
8    transition
9    1. State=0 /\ RCV(start) =|>
10   State':=1
11   /\ Nonce':=new()
12   /\ Request' := new()
13   /\ SND({Nonce'.Request'}_Ks)
14
15   2. State=1 /\ RCV({Nonce.Response'}_Kn) =|>
16   State':=2
17   %% Verify nonce
18   /\ request(N,S,auth_1,Nonce)
19   end role
20
21 role role_S(S:agent,N:agent,Kn:public_key,Ks:public_key,SND,RCV:channel(dy))
22   played_by S
23   def=
24   local
25   State:nat,Nonce:text,Request,Response:text
26   init
27   State := 0
28   transition
29   1. State=0 /\ RCV({Nonce'.Request'}_Ks) =|>
30   State':=1
31   /\ Response' := new()
32   /\ SND({Nonce'.Response'}_Kn)
33   /\ secret(Response',sec_1,{N,S})
34   /\ witness(S,N,auth_1,Nonce')
35   end role
36
37 role session(N:agent,S:agent,Kn:public_key,Ks:public_key)
38   def=
39   local
40   SND1,RCV1,SND2,RCV2:channel(dy)
41   composition
42   role_S(S,N,Kn,Ks,SND1,RCV1) /\ role_N(N,S,Kn,Ks,SND2,RCV2)
43   end role
44
45 role environment()
46   def=
47   const
48   kn,ks:public_key,
49   node,server:agent,
50   sec_1,auth_1:protocol_id
51   intruder_knowledge = {node,server,kn,ks}
52   composition
53   session(node,server,kn,ks)
54   end role
```

```
55
56 goal
57   secrecy_of sec_1
58   authentication_on auth_1
59 end goal
60
61 environment().
```

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
