# Peer review of "Decentralized Zone-Based PKI: A Lightweight Security Framework for IoT Ecosystems"

_information, doi:10.3390/info15060304_

Round 1
Reviewer 1 Report
Comments and Suggestions for Authors
The authors present a technical paper with relevant topic, proper research methodology and potentially good contribution to the field of studies.
The authors are encouraged to resubmit the paper with the required format and more mathematical clarity and system performance assessment metrics with the detailed discussion on the main experiment and possible application scenario with updated references with most recent results.
Author Response
Thank you very much for your valuable feedback on our paper. We truly appreciate your insights and suggestions for improvement.
In response to your comments, we have made significant enhancements to the mathematical clarity of our paper. Specifically, we have introduced several new formulas to provide a solid mathematical foundation for our performance analysis. Additionally, we have ensured that before presenting the performance tests, we clearly outline the metrics used for evaluation, providing readers with a comprehensive understanding of our methodology.
Furthermore, we have included a dedicated section discussing the possible applications of our proposed solution. This section delves into potential real-world scenarios where our solution can be applied, providing valuable context for readers and stakeholders.
Lastly, we have updated our references with the most recent results to ensure the integrity and relevance of our literature review.
We believe that these enhancements address the concerns raised in your review and significantly improve the overall quality and clarity of our paper. We are confident that these revisions will contribute to the advancement of the research in this field.
Once again, we appreciate your thoughtful feedback and look forward to resubmitting the revised paper for your consideration.
Reviewer 2 Report
Comments and Suggestions for Authors
The proposed article analyzes PKI lightweight security frameworks for IoT applications.
The work is well structured and very detailed (sometimes even too much, an overall reduced length of manuscript would be easier to approach for a reader.) I have only a few major remarks that I believe needs to be addressed before having the paper published:
1 - Sections such as 3.11 are tough to read, their structure is more like a report than a scientific paper. I suggest reorganizing them into much more compact tables.
2 -The paper's complexity and level of detail are not aligned with an overall poor state-of-the-art analysis. Section 2 could be enlarged with much more reference to current solutions. There is no detail on the overlaying HW capable of executing those algorithms, besides very general-purpose hardware. I'd consider adding a paragraph with details on the most relevant HW solutions for cryptography, and specify how they can relate to your work. non-exhaustive list of relevant work includes 10.1109/TC.2023.3278536, Rambus CMRT, ARM TrustZone, Intel SGX, AMD PSP.
Author Response
Thank you for your valuable comments and suggestions on our paper. We sincerely appreciate your insights, which have helped us to enhance the quality and clarity of our work.
In response to your feedback, we have made several improvements to address the concerns raised. Firstly, we have revised Section 3.11 to present the data in a more scientifically rigorous manner, utilizing compact tables to improve readability and adherence to the format of a scientific paper.
Additionally, we have revised and extended the introduction section to provide a more comprehensive explanation of the motivation behind our proposed solution. This revision aims to provide readers with a clearer understanding of the context and significance of our work.
While we have conducted an extensive systematic literature review (SLR) under review in another journal, we have added a most updated list of related works section to this paper, ensuring that readers have access to the latest research in the field.
Furthermore, we have addressed the lack of detail on the overlaying hardware capable of executing the algorithms proposed in our work. We have added a new section, titled "Hardware Considerations," to Section 2, providing an overview of the relevant hardware solutions for cryptography. This section includes details on specific hardware platforms such as Rambus CMRT, ARM TrustZone, Intel SGX, and AMD PSP, and highlights how they relate to our proposed solution.
Overall, we believe that these revisions significantly enhance the clarity, completeness, and relevance of our paper. We are confident that these improvements will address the concerns raised and contribute to the overall strength of our work.
Once again, we thank you for your thoughtful feedback and look forward to resubmitting the revised paper for your consideration.
Round 2
Reviewer 1 Report
Comments and Suggestions for Authors
The author presents a good technical paper with the relevant topic, proper research methodology and potentially good contribution to the field of studies. The author is encouraged to resubmit the final paper with the required format
Reviewer 2 Report
Comments and Suggestions for Authors
All my remarks have been addressed